# Inferior Mesenteric Artery Ligation Level in Rectal Cancer Surgery beyond Conventions: A Review

**DOI:** 10.3390/cancers16010072

**Published:** 2023-12-22

**Authors:** Antonio Brillantino, Jaroslaw Skokowski, Francesco A. Ciarleglio, Yogesh Vashist, Maurizio Grillo, Carmine Antropoli, Johnn Henry Herrera Kok, Vinicio Mosca, Raffaele De Luca, Karol Polom, Pasquale Talento, Luigi Marano

**Affiliations:** 1Department of Surgery, “A. Cardarelli” Hospital, Via A. Cardarelli 9, 80131 Naples, Italy; antonio.brillantino@gmail.com (A.B.); mauriziogrillomg@gmail.com (M.G.); 62antropoli@gmail.com (C.A.); 2Department of Medicine, Academy of Applied Medical and Social Sciences—AMiSNS: Akademia Medycznych I Spolecznych Nauk Stosowanych—2 Lotnicza Street, 82-300 Elbląg, Poland; j.skokowski@amisns.edu.pl (J.S.); k.polom@amisns.edu.pl (K.P.); 3Department of General Surgery and Surgical Oncology, “Saint Wojciech” Hospital, “Nicolaus Copernicus” Health Center, Jana Pawła II 50, 80-462 Gdańsk, Poland; 4Department of General Surgery and Hepato-Pancreato-Biliary (HPB) Unit—APSS, 38121 Trento, Italy; francesco.ciarleglio@apss.tn.it; 5Department Organ Transplant Center of Excellence, King Faisal Specialist Hospital and Research Center, Riyadh 11564, Saudi Arabia; yogesh.vashist@outlook.de; 6Department of General and Digestive Surgery—Upper GI Unit, University Hospital of León, 24008 León, Spain; drjhherrerak@gmail.com; 7Department of Advanced Medical and Surgical Sciences, Università degli Studi della Campania “Luigi Vanvitelli”, 80138 Napoli, Italy; vinicio.mosca@gmail.com; 8Department of Surgical Oncology, IRCCS Istituto Tumori “Giovanni Paolo II”, 70124 Bari, Italy; dr.raffaele.deluca@gmail.com; 9Department of Gastrointestinal Surgical Oncology, Greater Poland Cancer Centre, Garbary 15, 61-866 Poznan, Poland; 10Department of Surgery, Pelvic Floor Center, AUSL-IRCCS Reggio Emilia, 42122 Reggio Emilia, Italy; pastalento@gmail.com

**Keywords:** rectal cancer surgery, inferior mesenteric artery ligation, anastomoticleakage, quality of life

## Abstract

**Simple Summary:**

This research is dedicated to exploring the enduring discussion about the optimal level of ligation of the inferior mesenteric artery (IMA) during rectal cancer surgery, with an emphasis on historical, technical, and patient-centered dimensions. The study seeks to elucidate the unresolved issues by critically evaluating factors, such as the anastomotic leakage risk, genitourinary function implications, and oncological outcomes. The aim is to offer a nuanced perspective that transcends conventional paradigms, guiding surgeons and researchers toward a more individualized approach, mainly based on patient anatomy and surgeon preference. Efforts made by this research can lead to the refinement of surgical techniques and a better understanding of the intricate considerations involved in rectal cancer surgery, which can contribute to the ongoing evolution of medical practices in this area.

**Abstract:**

Within the intricate field of rectal cancer surgery, the contentious debate over the optimal level of ligation of the inferior mesenteric artery (IMA) persists as an ongoing discussion, influencing surgical approaches and patient outcomes. This narrative review incorporates historical perspectives, technical considerations, and functional as well as oncological outcomes, addressing key questions related to anastomotic leakage risks, genitourinary function, and oncological concerns, providing a more critical understanding of the well-known inconclusive evidence. Beyond the dichotomy of high versus low tie, it navigates the complexities of colorectal cancer surgery with a fresh perspective, posing a transformative question: “Is low tie ligation truly reproducible?” Considering a multidimensional approach that enhances patient outcomes by integrating the surgeon, patient, technique, and technology, instead of a rigid and categorical statement, we argued that a balanced response to this challenging question may require compromise.

## 1. Introduction

In the field of surgical oncology, rectal cancer surgery stands as a proof to the remarkable evolution of medical knowledge, surgical techniques, and therapeutic paradigms [1]. As we explore the intricacies of this complex and specialized surgery, it becomes evident that it is a continuously evolving field, deeply influenced by progresses in technologies and the extraordinary surgical intuitions of pioneering rectal surgeons [2]. 

The framework supporting modern therapeutic strategies for “high-quality” rectal surgery stands on the principles of “central devascularization” and “dissection within the embryological planes of the rectum and mesorectum” [3]. However, as we delve into this complex landscape, we deal with a persistent and unending debate regarding certain “details”. It is in this domain that the optimal level of the inferior mesenteric artery (IMA) emerges as an endless and passionate debate [4] in the early 20th century. In 1908, Miles WE et al. [5], who observed that patients with rectal cancers often exhibited proximal intra-abdominal lymph node involvement, introduced the abdominoperineal procedure. This innovative approach had a dual purpose: not only to excise the tumor but also to remove the lymphatic structures associated with cancer spread. It represented a groundbreaking advancement in surgical innovation. This technique involved the comprehensive transabdominal removal of lymphatic tissues and the ligation of the IMA, positioned just distal to the left colic branch, often referred to as the “low tie” technique. This approach represented a significant departure from the conventional surgical procedures of that time. 

In that same year, Moynihan BJ [6] proposed an alternative perspective, challenging the conventional knowledge regarding the ligation and division of the IMA. He concisely summarized his theory by stating, “We have not yet fully grasped that the surgery for malignant disease is not merely an operation on organs; it is the study of the lymphatic system”. In line with this perspective, Moynihan believed that ligating and dividing the IMA at the point where it connected flush with the aorta allowed for the removal of even more proximal lymph nodes, thereby reducing the risk of tumor recurrence. Subsequently, Dukes CE et al. [7] provided empirical evidence that the upward lymphatic extension of cancer consistently involved the lymph nodes closely associated with the IMA, extending right up to the aorta. This observation validated Moynihan’s “high tie” technique as a logical extension of radical rectal cancer excision. 

This historical review highlights the pioneering contributions of Miles, Moynihan, and Dukes, each of whom played a significant role in influencing the discussion concerning the ligation of the IMA in rectal cancer surgery. The implications of their insights continue to reverberate in contemporary surgical practices, underlining the enduring relevance of this unended debate. This controversy serves as the core focus of our narrative review. As we navigate through this intricate and multifaceted topic, we aim to provide an exhaustive exploration of the scientific context, the latest research, and the dynamic future directions surrounding this critical aspect of rectal cancer surgery.

## 2. Material and Methods

A literature search in the PubMed/Medline, Scopus, and Web of Science databases of all articles published until July 2023 was carried out. The combination of the following medical subject heading (MeSH) terms was used: “ligation” AND “high” OR “low” AND “inferior mesenteric artery” AND “surgery”. The key words were used in all possible combinations to retrieve the maximal number of articles. All types of study designs were included. Exclusion criteria included: articles in non-English language, experimental studies in animal models, abstracts, and editorials. Figure 1 illustrates the detailed steps followed during the literature search. The bibliography of each selected article was reviewed for other potentially relevant citations. 

## 3. Thirty-Nine mm: The Devil Is in the Details

In a study conducted by Murono et al. [8], the median length of the IMA from its origin to the take-off of its first branch was determined to be 39.3 mm, with a range spanning from 10.1 mm to 82.2 mm. This finding holds significant importance, as it is based on data gathered from 471 consecutive patients diagnosed with colorectal cancer who underwent preoperative 3D CT angiography. Therefore, it is fitting to emphasize the adage that “the devil is in the details,” as, within this relatively short 4 cm span along the IMA, numerous critical factors come into play. This constrained length raises concerns pertaining to blood supply, the risk of anastomotic leakage, oncological considerations, and functional outcomes. On the one hand, some surgeons advocate for high-tie ligation (HTL), prioritizing maximum lymph node clearance and facilitating the mobilization of the splenic flexure, sometimes at the potential cost of nerve injuries and compromised blood supply. In contrast, other authors recommend low-tie ligation (LTL), giving precedence to the preservation of intact blood supply and nerves, albeit with the possible consequence of suboptimal nodal clearance. Undoubtedly, the manipulation of the feeding vessels around the rectum has an immediate impact on vascularization, thereby influencing the viability of the neorectum during anastomotic healing. At the same time, it exerts a substantial influence on the extent of the central lymphadenectomy, with potential long-term oncological consequences [9]. The goals of blood vessel ligation in rectal cancer surgery are multifaceted and encompass achieving several crucial objectives. First and foremost, the primary aim is to ensure a comprehensive lymph node clearance. Simultaneously, it is imperative to preserve an adequate blood supply to the affected area and safeguard the integrity of adjacent structures, such as nerves. Moreover, a secondary objective involves facilitating the unrestricted mobilization of anastomotic stumps to enable the creation of a tension-free anastomosis. From a theoretical standpoint, it is worth noting that lymph node dissection should ideally be independent of the level of vascular ligation, provided that the vessels have been effectively cleared of lymphatic tissue for subsequent pathological analysis [10,11]. However, in real-world clinical scenarios, ligation at the level of the superior rectal artery does not necessarily mean the thorough removal of lymph nodes from the origins of the inferior mesenteric artery, the left colonic artery, and the superior rectal artery to achieve an adequate lymph node retrieval [10]. This highlights the practical challenges that surgeons encounter in ensuring a comprehensive lymphadenectomy during colorectal procedures. 

## 4. Keyword: Standardization

The historical uncertainty surrounding the precise location of the inferior mesenteric lymph nodes, particularly in relation to the IMA, has led to varying interpretations, describing this area both as the “root” and as part of the “periphery” [11,12]. To address this ambiguity and to promote uniformity in clinical practice, the Japanese Society for Cancer of the Colon and Rectum (JSCCR) has defined standardized criteria for classifying these nodes [13]. In accordance with the criteria established by the Japanese Society for Cancer of the Colon and Rectum (JSCCR), nodes situated at the origin of the inferior mesenteric artery (IMA) are categorized as station 253. This nodal grouping extends proximally along the IMA to the point where the left colic artery branches off. Conversely, nodes located within the inferior mesenteric trunk fall under station 252, encompassing the region extending from just distal to the origin of the left colic artery to the bifurcation point into the superior rectal artery. The standardization of these definitions serves to enhance precision in the execution and interpretation of research studies and significantly influences surgical decision-making. Within this standardized framework, the importance of surgical terminology becomes evident. High-tie ligation (HTL) entails ligating the IMA at its root, involving the dissection of nodes classified as station 253 (Figure 2). Conversely, LTL consists of ligating the IMA at or below the level of the left colic artery’s origin (Figure 3). In this approach, the primary aim is the removal of pericolic and intermediate groups of lymph nodes, including station 252 nodes. Similarly, on the western side, as reported in the Consensus Statement on Definitions for Anorectal Physiology and Rectal Cancer by the American Society of Colon and Rectal Surgeons (ASCRS), an LTL of the IMA is defined as a ligation below the origin of the LCA, while an HTL refers to a ligation at the point of origin from the aorta, preserving a 1.5–2 cm stump [14]. The standardization of these terms ensures that surgeons share a common language and understanding, ultimately leading to improvements in patient care and enhanced outcomes in colorectal surgery. 

## 5. The Surgeon- and Patient-Centric Approach: The Different Levels of the Debate 

Shifting the focus from a surgeon-centric approach, where we acknowledge the “technical level” of the debate, to patient-centered care, it is essential to recognize two additional distinct levels of consideration: the “quality of life (QoL) level” and the “oncological level”. Each of these dimensions plays a significant role in the decision-making process when it comes to vascular ligation. In order to provide a comprehensive overview, Table 1 reports outcomes of interest in patients receiving a high or low ligation of the IMA based on the selected literature.

### 5.1. Technical Level

Anastomotic leakage (AL) stands out as one of the most severe complications following rectal cancer surgery, leading to elevated rates of morbidity and mortality, resulting in prolonged hospitalization and increased medical costs [15,16]. Surgeons face the challenge of balancing technical considerations, such as ensuring adequate blood flow and minimizing tension at the anastomosis to reduce the risk of anastomotic leaks, all while adhering to oncologic principles. 

In the era of minimally invasive surgery, a growing preference has emerged among surgeons regarding the level of IMA ligation, with 91% opting for HTL, with only 9% choosing LTL [17,18,19,20,21,22,23,24,25,26,27,28]. This choice is frequently motivated by the straightforward execution of high ligation once access to the sigmoid mesentery has been established. When the IMA is ligated at its origin, the blood supply to the left colon relies on blood flow from the superior mesenteric artery via the middle colic artery, which provides the marginal artery of Drummond and the arc of Riolan [29,30]. However, it is noteworthy that Gourley and Gering [31] have reported the potential absence of the marginal artery in a subset of patients, ranging from 4% to 20%. For individuals with this anatomical variation, there is a theoretically increased risk of insufficient blood flow to the proximal part of the anastomosis, potentially leading to anastomotic leakage. Additionally, older individuals with atherosclerotic arteries and cardiovascular disease may experience a postsurgery drop in systemic blood pressure. In such cases, the blood pressure in the marginal artery may not be sufficient to sustain proper blood flow to the colon limb, even though the vascular system naturally regulates blood flow [29,32,33]. In these contexts, the ligation of the superior rectal artery (SRA) could represent a strategic and personalized approach to mitigate the inherent risk associated with these conditions. Existing studies have consistently revealed that there is no significant difference in AL rates between the use of HTL and LTL methods. A retrospective review conducted in Sweden from 2007 to 2009, involving 2023 rectal cancer patients, established that AL rates remained comparable irrespective of the ligation approach [34]. Moreover, a Japanese prospective randomized controlled trial, led by Fujii S. et al. [35], focusing on patients undergoing low anterior resection (LAR) for rectal cancer, indicated that the level of ligation did not determine a substantial impact on AL rates, although the trial was halted prematurely due to slowing enrollment. These findings were further substantiated in a systematic review conducted by Cirocchi R. et al. in 2012 [36], comparing IMA and SRA ligation methods. A recent meta-analysis by Yang Y et al. in 2018 [37] and Hajibandeh S. et al. [38] additionally affirmed the absence of significant differences in AL rates or the total number of harvested lymph nodes in patients undergoing rectal cancer surgery. The results from the large multi-institutional US Rectal Cancer Consortium (USRCC), on 877 rectal cancer patients who underwent LAR or abdominoperineal resection (APR), confirmed that the ligation method was not associated with AL (OR = 1.82 (95% CI, 0.48–6.92); *p* = 0.38), contrary to female sex and smoking history [17]. Nonetheless, there had been no clear explanations to support those results, partly due to certain limitations in the study. For instance, the inclusion of patients treated with both LAR and APR collectively might have contributed to these shortcomings.

In recent years, there has been a growing interest in utilizing infrared fluorescence imaging with indocyanine green (ICG) to assess the perfusion status in colorectal surgery, aimed at addressing an ongoing debate. Several research studies have documented that the use of fluorescent angiography with ICG may potentially reduce the rate of AL [39,40,41,42]. Moreover, recent investigations have conducted quantitative analysis of ICG fluorescent imaging, highlighting its correlation with the risk of anastomosis leakage [43,44]. Intriguingly, in a randomized controlled trial [15], researchers conducted a comparative analysis of the colonic perfusion status between the HTL and LTL groups using a quantitative near-infrared (NIR)–ICG fluorescence perfusion test. The study’s findings revealed that there was no significant difference in the perfusion intensity, particularly in the F_max, which represents the fluorescence difference between the maximum and baseline intensity, between the HL and LL groups (0.76 ± 0.27 vs. 0.80 ± 0.26; *p* = 0.768, respectively). This suggests that, after HTL, there may be a delay in perfusion due to the blood supply from the right side, but the overall intensity and quantity of the perfusion do not seem to objectively depend on the level of IMA ligation. When considering the surgical technique, it becomes evident that HTL involves removing a substantial portion of the sigmoid colon during rectal resection, necessitating the use of the descending colon for anastomosis. Given this context, it is imperative to underscore the significance of yet another crucial maneuver, which is the ligation of the inferior mesenteric vein (IMV). It is a key factor for ensuring a tension-free creation of a low colorectal or coloanal anastomosis [45]. Scientific evidence is supported by a study conducted on 13 adult fresh cadavers showing that the increase in the colon length following IMV ligation exceeded the length obtained with low IMA ligation, high IMA ligation, and high IMA ligation in combination with splenic flexure mobilization (*p* < 0.0001) [46]. Similarly, Girard E et al. [45] also demonstrated, on embalmed anatomical specimens, that, in 80% of cases, arterial ligation did not allow length gain when not accompanied by the IMV section, regardless of the level. Several authors have attempted to also assign an oncological role to this procedure, asserting that it can substantially increase the number of resected lymph nodes along the root of the IMV, potentially influencing both the survival and the local recurrence rate [47,48]. However, the published data have shown contrasting results, likely due to the heterogeneity in the patient population, leaving the oncological significance of the IMV ligation uncertain [49,50]. Only one recent multicenter randomized controlled trial explored the effects of an extended lymphadenectomy in sigmoid colon cancer [50]. Their findings indicated that including the lymphatic area around the IMV did not result in a higher total node count or a greater detection of positive nodes. Furthermore, it did not lead to improvements in disease-free survival (DFS) or overall survival (OS). Notably, the median number of lymph nodes extracted from the additional tissue surrounding the IMV in the extended lymphadenectomy group was just one. In conclusion, it seems that the HTL of the IMA is primarily justified from a technical perspective for surgeons and is predominantly based on non-oncological considerations. The HTL of the IMA, when combined with the high ligation of the IMV, can be performed safely and, in comparison to LTL, offers an additional length that can be essential for achieving a tension-free low colorectal or coloanal anastomosis.

### 5.2. Quality of Life Level

In the contemporary healthcare landscape, we are observing a shift away from the traditional disease-focused model to a patient-centered approach [51]. This transition emphasizes the growing importance of patient-centered assessments, which include key measures like QoL as integral indicators of intervention outcomes. This multidimensional assessment becomes even more relevant in the context of rectal cancer surgery, given the distinct challenges posed by its anatomical location, including issues related to exposure in a narrow pelvis, low intestinal transection, total mesorectal excision (TME), and the intricacies of nerve-sparing techniques [52,53]. Furthermore, these distinctive complexities make it challenging to fully understand the specific impact of the arterial ligation level on the genitourinary (GU) function [54,55,56,57]. The para-aortic trunks are derived from the mesenteric plexus and follow a descending course along the aorta, ultimately converging to constitute the superior hypogastric plexus. Hence, in the cases of HTL, it becomes imperative to meticulously identify the optimal ligation point of the IMA to mitigate the risk of autonomic nerve injury during rectal cancer surgery [57]. Notwithstanding, post rectal surgery urinary dysfunctions are documented to affect up to 30% of patients, and over half of them also experience sexual dysfunction [58,59]. These issues, even if not often discussed in clinical practice, collectively contribute to a negative impact on the patients’ quality of life [58]. From an anatomical standpoint, there are ongoing divergences in perspectives concerning the relationship between the chosen level of IMA ligation and the resulting impact on autonomous nerve integrity. While some authors assert that the safest point for ligation of the IMA is its origin, others argue that, due to the complex and intricate extension of the inferior mesenteric plexus around the IMA’s origin, and for the initial tract from the aortic plane, an HTL might more easily lead to sympathetic nerve injury [60,61,62]. From a clinical perspective, the literature presents an even more confusing picture. Previous studies investigating GU function outcomes have not definitively demonstrated the superiority of a specific approach to IMA ligation, primarily due to the absence of well-designed RCTs, the heterogeneity found in retrospective reports, and the reliance on patient questionnaires as the sole method of assessment [28,29,36,63,64,65]. In their 2012 meta-analysis, Cirocchi et al. [36] compared HTL versus LTL in LAR, revealing the need for a well-designed randomized controlled trial (RCT) to further clarify the outcomes. Subsequently, in a more recent study, they found similar postoperative urinary retention rates between the two ligation groups, emphasizing that such retention is typically not nerve-related but rather associated with bladder sphincter paralysis [4]. Surprisingly, the analysis indicated a favorable postoperative urinary incontinence rate in the HTL group, while the LTL group exhibited better preservation of male sexual function. Notably, the study’s limitation lies in a small sample size, and the lack of data on QoL outcomes for female patients undergoing rectal resection with different ligation techniques underscores the need for further research. Mari G et al. [56] aimed to address the literature gap with the HIGHLOW trial, an RCT comparing the incidence of GU dysfunction through standardized survey evaluation, and uroflowmetric examination in 214 patients undergoing elective laparoscopic LAR and TME. The LTL group exhibited significant postoperative improvements, reporting enhanced continence, reduced obstructive urinary symptoms, and an overall improved QoL. Additionally, sexual function was better preserved in the LTL group compared to the HTL group at the same postoperative interval. While these findings suggest that the LTL technique is associated with superior GU and sexual function outcomes, the study acknowledges limitations, such as the nonhomogeneous tumor stage distribution, inadequate statistical power for female patients, and challenges posed by the small sample sizes. Consequently, cautious interpretation is advised, and the authors emphasize the imperative for further research to address these limitations and provide a more comprehensive understanding of the observed outcomes. The regulation of the GU function is intricately influenced by several factors, including the posterior tilting of the bladder following rectal surgery, inflammation of paravesical tissues, neoadjuvant chemoradiation therapies, tumor size, and the presence of lymph node metastases [58,59,66,67]. This complexity poses challenges in isolating and precisely evaluating the specific determinants impacting the GU function, although intraoperative injury to pelvic autonomic nerves emerges as a primary causative factor. Moreover, pelvic nerve injuries may manifest at different stages throughout LAR with TME, encompassing not only the critical point of IMA ligation but also specific pivotal moments within TME and during perineal dissection [68]. This underscores the significance of considering the entire surgical procedure in assessing the risk and occurrence of pelvic nerve injuries. While the dissection of pelvic nerves presents technical challenges, the synergistic application of minimally invasive techniques, fluorescence-guided surgery utilizing dedicated immunostaining, and computerized imaging collectively enhances the delineation of pelvic organs and clarifies their innervation [69]. The incorporation of three-dimensional (3D) models facilitates a more comprehensive understanding of complex anatomic structures [70,71].

### 5.3. Oncological Level

Over the years, another ongoing debate has centered around the question of whether to tie the IMA at its origin or to opt for ligating the SRA while preserving the left colic artery to effectively address oncological concerns [72,73,74,75,76,77,78,79,80,81]. The prospective (but uncontrolled) study conducted by Kanemitsu Y et al. [11] on 1188 patients who underwent curative treatment for rectal and rectosigmoid junction cancer with HTL of the IMA epitomizes the application of standardized criteria in accordance with JSCCR’s classification to define the location of inferior mesenteric lymph nodes. The primary objective of their study was to assess the potential survival advantages of HTL in patients with metastatic involvement in specific lymph nodes that would be left behind after LTL procedures. The incidence of metastasis to station 253 nodes, representing residual metastatic nodes that would typically be left behind in an LTL, was notably low at 1.7% (20 of 1188). These patients exhibited 5-year overall and cancer-specific survival rates of 40% and 42%, respectively, indicating an enhancement in the long-term survival compared to LTL. Although the routine use of HTL during curative resections showed relatively modest benefit, with an incidence rate of metastasis at 1.7% and an estimated 0.7% of patients likely to achieve cure through HTL of the IMA, the rationale behind HTL remained intact. Additionally, the negligible operative mortality (0.2%) and morbidity rates (31.5%) confirmed the safety of HTL, reinforcing its applicability in cases where it aligns with the patient’s best management [82]. A large Korean retrospective study on 1213 patients who underwent LAR for stage I to III rectal cancer reported similar 5-year locoregional recurrence-free survival (RFS) (92% vs. 96%; *p* = 0.20) and 5-year OS (88% vs. 93%; *p* = 0.17) for patients who underwent the high- versus low-tie approach [83]. From the western perspective, the US Rectal Cancer Consortium (USRCC) examined the impact of IMA ligation on oncologic outcomes on a total of 877 rectal cancer patients undergoing either LAR or APR. The two different methods, HTL and LTL, showed equivalent results in terms of lymph node harvest adequacy (median, 15 (12–19) vs. 15 (13–20), *p* = 0.38), locoregional 5-year RFS (87% vs. 90%; *p* = 0.456), and 5-year RFS (85% vs. 82%; *p* = 0.326), respectively [17]. More recently, Mari G et al. [84] published updated results from the HIGHLOW trial, aimed at reporting the 5-year oncologic outcomes of 196 patients who underwent LAR + TME with either an HTL or LTL of the IMA. The study found no significant differences in the distant recurrence rate (15.8% vs. 18.9%; *p* = 0.970), pelvic recurrence rate (4.9% vs. 3.2%; *p* = 0.843), 5-year OS (80.8% vs. 81.9%; *p* = 0.545), and DFS (79.2% vs. 78%; *p* = 0.985) between the HTL and LTL, respectively [84]. Despite its randomized controlled design, the study exhibits several limitations, particularly in terms of the potential influence of preoperative treatments and the precise documentation of the locations of local or distant recurrence within both the HTL and LTL groups [85]. Additionally, since the sample size calculation was primarily tailored for evaluating GU rather than oncological outcomes, the study may be underpowered. The imperative for an ad hoc randomized study, with a larger sample size tailored on the oncological outcomes, coupled with the need to address intersurgeon variabilities, is evident for achieving statistically significant results in this context. Other studies have explored this research question, and despite being summarized in various systematic reviews and meta-analyses, a conclusive answer is still elusive [4,35,36,37,86,87,88,89,90,91,92,93,94]. Moreover, primary research studies face various limiting factors, including missing data, a low sample size, and inadequate comparisons that combine rectosigmoid cancers with true rectal cancers. Additionally, there may be issues with the assessment and follow-up procedures. Consequently, it is not surprising that systematic reviews and meta-analyses produce conflicting results. One may not detect significant differences in oncological outcomes while another, published a year later, may show the opposite. While systematic reviews and meta-analyses are considered the apex of the pyramid of the evidence hierarchy, their reliability is compromised if they are based on weak foundations; namely, primary studies. Just like the most perfectly structured geometrical shape will collapse if its underlying pillars are inherently weak, the validity of these reviews and analyses is jeopardized when built upon less-robust primary research. For these reasons, another systematic review or meta-analysis would not be able to add anything more to the current literature. Anyway, one of the most meticulously designed and executed studies, particularly from a methodological standpoint, rarely cited and included in systematic reviews and meta-analyses, is a retrospective Swedish population-based cohort study [95]. This study aimed to investigate the impact of high tie on the survival and cancer recurrence after rectal cancer surgery on a total of 8287 patients who underwent rectal cancer resection in Sweden from 2007 to 2014. It employed contemporary epidemiological and statistical methods, along with a sufficiently large sample size to draw conclusions even from negative results. Eligible patients were identified from the Swedish Colorectal Cancer Registry, known for its >97% coverage of rectal cancer patients in Sweden and its comprehensive information on patient characteristics, intraoperative data, postoperative course, pathological assessment, oncological treatment, and other relevant data [96]. After propensity score matching, performed to adjust for potential confounding factors and emulate a randomized trial, 2907 patients were selected for each group. Interestingly, there was no association found between the level of the tie and locoregional (HR 0.85, 95% CI 0.59–1.23) or distant (HR 1.01, 95% CI 0.88–1.15) recurrence, nor with cancer-specific (HR 0.92, 95% CI 0.79–1.07) or overall (HR 0.98, 95% CI 0.89–1.08) survival. Additionally, adjuvant chemotherapy did not significantly affect the result regarding cancer-specific survival (HR 0.93, 95% CI 0.81–1.08), suggesting no residual confounding. Surprisingly, already 40 years ago in 1984, Nicholls J. obtained similar results, asserting that HTL of the IMA does not enhance the survival of patients with rectal and rectosigmoid cancer [53]. The conclusion of that study, stating “While most surgeons performing these operations tended to favor one type of ligation over the other, the possibility of some selection cannot be ruled out…”, seems to foreshadow the conclusions stated by Boström P et al. in 2019, “…the level of vascular tie can be determined by the patient’s anatomical configuration and the surgeon’s preference, rather than on oncological concerns”. In this regard, when considering the multitude of interindividual variations in the anatomy of the IMA’s division branches, it becomes challenging [97,98,99], if not nearly impossible in some cases, to execute an LTL [100,101]. To navigate this challenge, the use of preoperative CT angiography proves invaluable in assessing the anatomical intricacies of the IMA and its branches. This preoperative evaluation equips surgeons with essential insights, enabling the tailored planning of surgical interventions to suit the unique anatomical variations observed in individual patients [100,101]. Once again, from a technical standpoint, HTL emerges as a more feasible option, representing the reproducible and standardizable technique applicable to all patients. 

**Table 1 cancers-16-00072-t001:** Selected literature: outcomes in patients with high or low IMA ligation.

Study	Year	Design	Tumor Location	Surgical Approach	Inferior Mesenteric Artery Ligation Level		Overall Complication Rate (%)		Mortality Rate (%)		Anastomotic Leakage (%)		5-Year OS (%)		5-Year RFS (%)	
					HTL	LTL	HTL	LTL	HTL	LTL	HTL	LTL	HTL	LTL	HTL	LTL
AlSuhaimi MA et al. [83]	2019	Non-RCT	Rectal cancer	Laparoscopic > Robotic > Open	835	378	11.%	10.8%	-	-	4.8%	3.2%	75.4%	80.6%	92.1%	96.1%
Bostrom P et al. [95]	2019	Non-RCT (PSM)	Rectal cancer	Open > Laparoscopic	2907	2907	-	-	2.1%	2.4%	-	-	76.5%	75.9%	-	-
Chen JN et al. [20]	2020	Non-RCT	Rectal cancer	Laparoscopic	235	237	-	-	-	-	11%	2.8%	-	-	-	-
Dimitriou N et al. [19]	2018	Non-RCT	Sigmoid/Rectal cancer	Open	76	44	6.5%	25%	3%	5%	1%	5%	-	-	-	-
Draginov A et al. [21]	2020	Non-RCT	Rectal/Rectosigmoid cancer	Laparoscopic > Robotic	158	123	24.7%	34.1%	0	0	3.2%	5.7%	-	-	-	-
Fujii S et al. [35]	2018	RCT	Rectal cancer	Open > Laparoscopic	164	160	37.2%	35%	0	0.6%	17.7%	16.3%	87.2%	89.4%	76.3%	77.6%
Gömcel L et al. [22]	2020	Non-RCT	Rectal cancer	Robotic	39	38	30.8%	18.4%	-	-	7.7%	5.3%	-	-	-	-
Guo Y et al. [23]	2015	RCT	Rectal cancer	Laparoscopic	29	28	-	-	-	-	10.3%	3.6%	-	-	-	-
Hinoi T et al. [24]	2013	Non-RCT	Rectal cancer	Laparoscopic	304	584	22.7%	22.3%	-	-	13.2%	7.4%	-	-	-	-
Hsu CC et al. [25]	2023	Non-RCT (PSM)	Sigmoid/Rectal cancer	Laparoscopic > Open > Robotic	245	245	13.1%	7.4%	0	0.8%	5.7%	3.3%	-	-	-	-
Hu S et al. [26]	2021	Non-RCT	Rectal cancer	-	65	75	7%	9%	-	-	3%	4%	88.2% (3-y)	97% (3-y)	86.1% (3-Y)	83.9% (3-y)
Kim CS et al. [27]	2019	Non-RCT	Sigmoid/Rectal cancer	Laparoscopic > Open > Robotic	97	97	22.7%	30.9%	-	-	14.4%	5.2%	-	-	-	-
Komen N et al. [28]	2011	Non-RCT	Rectal cancer	Open	16	17	-	-	-	-	0.06%	0.06%	-	-	-	-
Kruszewski WJ et al. [93]	2021	RCT	Rectal/Rectosigmoid cancer	Open	65	65	63%	57%	-	-	3%	5%	77%	76%	72%	81%
Lee KH et al. [72]	2018	Non-RCT	Sigmoid/Rectal cancer	Laparoscopic	51	83	41.1%	45.7%	-	-	3.9%	0	84.1%	87.5%	92.6%	91.1%
Luo Y et al. [73]	2021	Non-RCT	Rectal cancer	Open	295	221	38.3%	41.2%	1.4%	0.9%	13.2%	8.6%	69.1%	69.6%	56.2%	59.5%
Luo Y et al. [74]	2021	Non-RCT	Rectal cancer	Laparoscopic	378	236	35.9%	30%	0	0	14.3%	8.9%	61.4%	69.5%	53.5%	52.2%
Mari G et al. [56,84]	2018	RCT	Rectal cancer	Laparoscopic	101	95	27.9%	30%	-	-	8.1%	6.7%	80.8%	81.9%	79.2%	78%
Matsuda K et al. [75]	2015	RCT	Rectal cancer	Laparoscopic > Open	51	49	35.3%	32.6%	0	0	16%	10%	-	-	-	-
Nayery M et al. [76]	2019	Non-RCT	Rectal cancer	Laparoscopic	101	99	43.5%	30.3%	0	0	23.7%	15.1%	90.1%	92.9%	83.2%	77.8%
Park SS et al. [77]	2020	Non-RCT	Sigmoid/Rectal cancer	Laparoscopic	613	163	25.1%	28.8%	-	-	2.8%	2.5%	79.6%	81.3%	77.4%	73.3%
Pezim ME et al. [53]	1983	Non-RCT	Rectal cancer	Open	586	784	64%	63.5%	3.1%	2.2%	-	-	59.7%	57.4%	-	-
Turgeon MK et al. [17]	2021	Non-RCT	Rectal cancer	Laparoscopic > Open	755	122	51%	51%	-	-	8%	9%	90%	85%	85%	82%
Yamamoto M et al. [78]	2014	Non-RCT	Sigmoid/Rectosigmoid cancer	Laparoscopic	100	181	12%	10.8%	-	-	2.2%	1.6%	94.8%	91.8%	93%	87.6%
Yasuda K et al. [79]	2016	Non-RCT	Sigmoid/Rectal cancer	Open	42	147	19%	17%	-	-	4.8%	2%	82.4%	80.3%	75.6%	76.2%
You X et al. [81]	2020	Non-RCT	Rectal cancer	Laparoscopic	174	148	-	-	0	0	9.7%	3.4%	77%	77%	-	-
Yu J et al. [94]	2022	Non-RCT	Rectal cancer	Laparoscopic	134	86	25.4%	20.9%	1.5%	1.2%	10.4%	8.1%	78.3%	82.4%	72.4%	76.6%
Zhang C et al. [80]	2020	Non-RCT	Rectal cancer	Laparoscopic	126	79	-	-	-	-	1.6%	0	78.1%	87.7%	-	-

RCT, randomized controlled trial; PSM, propensity score matching; HTL, high-tie ligation; LTL, low-tie ligation; OS, overall survival; RFS, recurrence-free survival.

## 6. Conclusions

In this comprehensive narrative review exploring the optimal level of IMA ligation in rectal cancer surgery, our investigation explored the historical origins, technical considerations, and outcomes across various levels. The analysis was conducted through the lenses of technical perspectives, QoL implications, and oncological viewpoints. At the technical level, the debate centers on the AL risk, with studies suggesting no significant difference between high- and low-tie approaches. The QoL aspect draws attention to the GU function, where the evidence, although inconclusive, hints at potential benefits associated with LTL. Finally, the oncological dimension scrutinizes the survival and recurrence outcomes, emphasizing a lack of substantial differences between the two ligation methods. Interestingly, a recent large-scale study, employing rigorous propensity score matching, found no association between the level of the tie and locoregional or distant recurrence, and cancer-specific or overall survival, challenging long-standing assumptions. This review significantly contributes to the ongoing debate, emphasizing the need for individualized approaches influenced by patient anatomy and surgeon preference, rather than strict adherence to oncological concerns. It is time to move beyond the conventional strategies to avoid encountering the same inconclusive results. A shift in perspective is necessary. Instead of framing the question as ‘Is HTL better than LTL?’, a more pertinent query is “Is LTL truly reproducible in colorectal cancer surgery?” A balanced response to this challenging question could involve compromise. Surgeons and researchers should imagine a rectangle where the “surgeon,” “patient,” “technique,” and “technology” angles are interconnected and converge at the center. This central point represents an optimal outcome for the patient, where all these elements work cohesively.

## Figures and Tables

**Figure 1 cancers-16-00072-f001:**
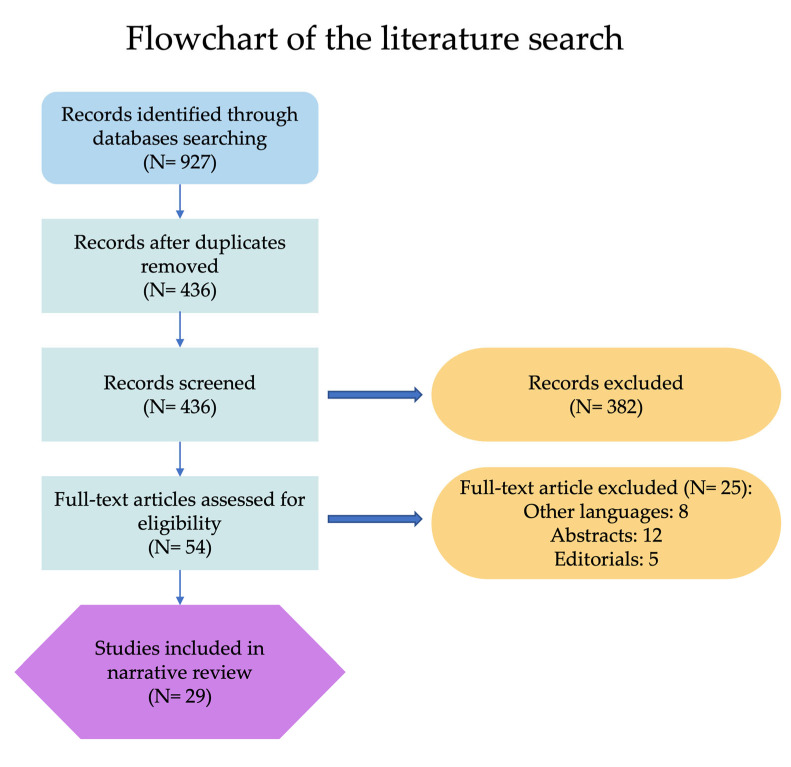
Flow diagram showing the process of the literature search.

**Figure 2 cancers-16-00072-f002:**
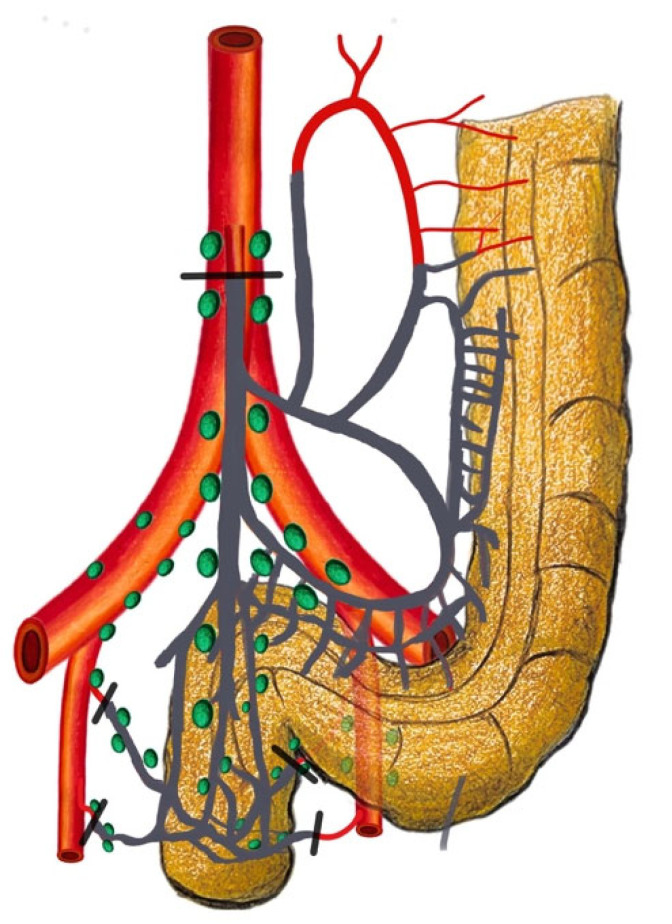
High-tie ligation: ligation of the inferior mesenteric artery at its origin with left colic artery ligation.

**Figure 3 cancers-16-00072-f003:**
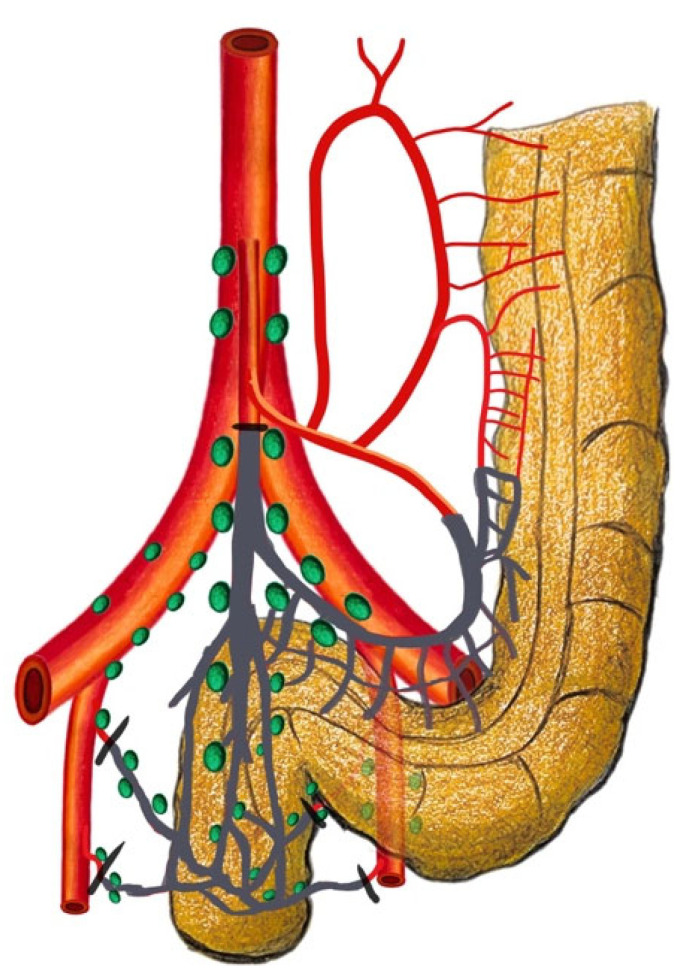
Low-tie ligation: ligation of the superior rectal artery just distal to the left colic artery’s origin.

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
