# Peer review of "Inferior Mesenteric Artery Ligation Level in Rectal Cancer Surgery beyond Conventions: A Review"

_cancers, 2023, doi:10.3390/cancers16010072_

Round 1

Reviewer 1 Report

Comments and Suggestions for Authors

Exhaustive literature review for a subject often studied but without clear conclusion

The 2 figures showing the ligation levels are clear, a third option should be discussed: Ligation of sigmoidal trunk after the departure of left superior colic artery (technique that we realize)

The two mains objective of this paper should be to conclude advantages of each technique for Perfusion of the colon and length of the colon. Rightly, the problem of IMV is crucial

The oncology chapter should be more succinct by omitting the old studies. Oncologic results are dependants of the initial stage, neo adjuvant treatments and tumor response.

Reviewer 2 Report

Comments and Suggestions for Authors

thank you for allowing me to review this literature on the level of section of the inferior mesenteric artery. the manuscript is well written and attempts to answer 3 questions. what is the impact of the level of section of the inferior mesenteric artery on the prevalence of: anastomotic fistulas; urinary sequelae; and oncologic outcome.

in the chapter on material and methods, i suggest that the authors add a paragraph on the methodology used to select the articles: inclusion and exclusion criteria; years of collection. in the chapter on results, a flow chart will show the number of articles selected.

for greater visibility, i suggest that authors add literature tables to illustrate the results, thus limiting the length of the text.

the authors cite a study on the impact of anastomotic fistulas, but include both anastomoses and abdominoperineal amputations (reference 17). we fail to see the relevance of the level of AMI section in abdominoperineal amputation in terms of anastomotic fistulas.

Similarly, when assessing the impact of vascular ligation on the prevalence of anastomotic fistula, homogeneous patient groups must be compared: neoadjuvant or non-neoadjuvant radiotherapy, lowering or not of the left colonic angle, section at the origin of the inferior mesenteric vein, type of anastomosis (termino-terminal; latero-terminal, J-shaped reservoir) and finally height of the anastomosis: factors that may influence the prevalence of anastomotic fistula.

these findings also apply to urinary function sequelae.

finally, for carcinological outcome, the only relevant study that should appear first is that from the Swedish registry, where the use of a propensity score makes it possible to homogenize patient groups.

finally, the level of ligation of the lower mesenteric artery seems to have little influence on the prevalence of anastomotic fistulas, urinary sequelae and carcinological results. it seems interesting in the perspective to discuss the place :  (i) of the abdominopelvic angioscanner in order to visualize the arterial anatomical variations; (ii) the potential impact of fluorescence which may or may not intersect the colon to prevent the vascular defect of the lowered colon

Round 2

Reviewer 2 Report

Comments and Suggestions for Authors

the authors have responded point by point to comments and questions designed to improve the quality of the manuscript